



# 1 Above- and Belowground Plant Mercury Dynamics in a Salt Marsh
# 2 Estuary in Massachusetts, USA

Ting Wang[1], Buyun Du[1], Inke Forbrich[2], Jun Zhou[1], Joshua Polen[1], Elsie M. Sunderland[3], Prentiss H.
Balcom[3], Celia Chen[4], Daniel Obrist[1,5*]
[1]Department of Environmental, Earth, and Atmospheric Sciences, University of Massachusetts Lowell, Lowell, MA 01854, USA
[2]Marine Biological Laboratory, Woods Hole, MA 02543, USA
[3]Harvard John A. Paulson School of Engineering and Applied Sciences, Harvard University, Cambridge, MA 02138, USA
[4]Department of Biological Sciences, Dartmouth College, Hanover, NH 03755, USA
[5]Division of Agriculture and Natural Resources, University of California, Davis, CA 95618, USA
*Correspondence to:* Daniel Obrist (daniel_obrist@uml.edu)
**Abstract.** Estuaries are dominant conduits of mercury (Hg) to the coastal ocean and the salt marshes within play an important role
in coastal Hg cycling. While Hg cycling in upland terrestrial systems has been well studied, processes in salt marsh ecosystems
are poorly characterized. We investigated Hg dynamics in vegetation and soils in the Plum Island Sound estuary in Massachusetts,
USA and specifically assessed the role of marsh vegetation for Hg deposition and turnover. Monthly quantitative harvesting of
aboveground biomass showed strong linear seasonal increases in plant Hg, with a four-fold increase in Hg concentration and an
eight-fold increase in standing Hg mass between June ($3.9\pm0.2$ µg kg$^{-1}$ and $0.7\pm0.4$ µg m$^{-2}$, respectively) and November ($16.2\pm2.0$
µg kg$^{-1}$ and $5.7\pm2.1$ µg m$^{-2}$, respectively). Hg ceased to increase in aboveground biomass after plant senescence, indicating
physiological controls of vegetation Hg uptake in salt marsh plants. Hg concentrations in live roots and live rhizomes were 11
times and two times higher than concentrations in aboveground live biomass, respectively. Furthermore, live belowground biomass
Hg pools (roots and rhizomes, $108.1\pm83.4$ µg m$^{-2}$) is more than ten times larger than peak standing aboveground Hg pools ($9.0\pm3.3$
µg m$^{-2}$).
A ternary mixing model suggests Hg sources in marsh aboveground tissues originates from a mix of root uptake (~35%),
precipitation uptake (~33%), and atmospheric gaseous elemental mercury (GEM) uptake (~32%). The results suggest a more
important role of Hg transport from belowground (i.e., roots) to aboveground tissues in salt marsh vegetation compared to upland
vegetation, where GEM uptake is generally the dominant Hg source. GEM deposition via uptake and subsequent senescence (5.9
µg m$^{-2}$ yr$^{-1}$) and throughfall (1.0 µg m$^{-2}$ yr$^{-1}$) hence is lower in this salt marsh ecosystem compared to upland vegetation and is
similar to open field wet and dry deposition (6.2 µg m$^{-2}$ yr$^{-1}$). Hg contained in salt marsh aboveground tissues leads to direct Hg
export to tidal water and oceans via wrack (tidal flushing of vegetation), which accounts for ~1.6 µg m$^{-2}$ yr$^{-1}$. Hg consumption by
herbivory ranges between 0.5 and 2.4 µg Hg m$^{-2}$ yr$^{-1}$. The similarity in isotopic signatures between roots and soils suggest that
belowground plant tissues mostly take up Hg directly from soils. Annual root turnover results in large internal Hg recycling
between soils and plants accounting for 58.6 µg m$^{-2}$ yr$^{-1}$. An initial mass balance of Hg in this whole estuarine salt marsh ecosystem
considering atmospheric inputs (atmospheric GEM and precipitation Hg(II), throughfall, including plants) and losses (wrack export
and lateral exchange of dissolved and particulate Hg) shows that the salt marsh presently serves as a small net Hg sink for
environmental Hg of 5.2 µg m$^{-2}$ yr$^{-1}$.





## 1. Introduction

Coastal salt marshes are ecosystems located at the interface between terrestrial and marine ecosystems and experiencing twice daily saltwater inundation by tidal water. They provide important ecological services, have high socioeconomic benefits, and serve as sinks and sources of carbon, nutrients, and contaminants (Hopkinson et al., 2018; Morris et al., 2013). Riverine export is the largest source of mercury (Hg) to coastal oceans globally (Amos et al., 2014; Liu et al., 2021). The location of salt marshes at this interface merits an understanding of their respective Hg sinks and sources and role in coastal Hg cycling. The Plum Island Sound salt marsh in Massachusetts, USA is the largest macrotidal marsh estuary in New England and is considered a biological mercury (Hg) hotspot, with 62% of saltmarsh sparrows reportedly exceeding a blood Hg threshold that may reduce nesting success (Evers et al., 2007; Jackson et al., 2011; Lane et al., 2020; Lane et al., 2011). While there is no direct evidence of particular Hg point sources within this watershed (Wang and Obrist, 2022), possible sources in the salt marsh estuary include atmospheric deposition directly to the marsh and its watershed (Evers et al., 2007; Lane et al., 2011). A recent investigation of the salt marsh indicated high Hg concentrations in this marsh soils and showed evidence that this salt marsh ecosystem currently serves as a net source of Hg via lateral tidal Hg export to tidal water and the ocean (Wang and Obrist, 2022).

A potentially important Hg source in marshes also includes Hg uptake by plants. In terrestrial environments, plants assimilate substantial amounts of atmospheric Hg, which is subsequently transferred to soils via tissue senescence (e.g., litterfall) and wash-off (i.e., throughfall deposition; review by Zhou et al., 2021). Plant Hg uptake is generally dominated by assimilation of atmospheric gaseous elemental Hg (GEM), and global vegetation acts as a large atmospheric GEM pump to soils (Jiskra et al., 2018; Obrist et al., 2018; Zhou et al., 2021; Zhou and Obrist, 2021). In terrestrial ecosystems, Hg inputs derived from plants are the dominant Hg sources accounting for 60% to 90% of total Hg inputs (Zhou and Obrist, 2021). Salt marshes are characterized by high plant net primary productivity (NPP) driven by vascular macrophytes, with plant NPP as high and even exceeding that of terrestrial ecosystems (Marques et al., 2011;Tobias and Neubauer, 2009; Visser et al., 2018). For example, salt marsh biomass production across Atlantic and Gulf sites in the U.S. ranges from 228 to 1,335 g C $m^{-2}$ $yr^{-1}$ with a median value of 537 g C $m^{-2}$ $yr^{-1}$ (Tobias and Neubauer, 2009). By comparison, NPP across 18 productive U.S. forests ranging between 400 to 1,000 g C $m^{-2}$ $y^{-1}$ (He et al., 2012). As a result, salt marshes are considered strong sinks of atmospheric carbon driven by plant $CO_2$ assimilation (Forbrich et al., 2018).

We hypothesized that salt marsh plants in the Plum Island estuary salt marsh act as substantial sinks of atmospheric Hg via vegetative assimilation of GEM, and aimed to quantify Hg sources in salt marsh vegetation, accumulation rates, and turnover rates of Hg in salt marsh plants. The specific objectives of this study were to quantify: (i) Hg fluxes and pools associated with plant dynamics in the salt marsh and Hg associated with annual growth of aboveground tissues; (ii) transfer of Hg associate with aboveground tissues to soils during senescence; (iii) Hg turnover in belowground biomass, a potentially important flux given that belowground biomass production in salt marshes is equal to or greater than aboveground production (Blum, 1993; Morris, 2007); (iv) specific sources of Hg in salt marsh biomass tissues using stable Hg isotope signatures to evaluate the implications of biomass dynamics for salt marsh Hg cycling and atmospheric Hg deposition; and (v) an initial ecosystem Hg mass balance in this salt marsh based on our observations together with other available data sources.

## 2. Method

### 2.1. Site Description

Sampling sites WERE located in the Plum Island Sound on the northeastern coast of Massachusetts, USA (42°45'10", 70°56'46") between the Gulf of Maine and the city of Boston. The estuary is the largest marsh-dominated estuary in New England with a total



marsh area of 60 km$^2$ and salt marsh area of 40 km$^2$ (Hopkinson et al., 2018; Millette et al., 2010). Tides are semidiurnal with an
amplitude averaging 2.7 m (NOAA Tide Predictions, 2020). We focused our study on high marsh platforms, with an approximate
elevation of 1.4 m above the North American Vertical Datum 88, which dominate tidal marshes in New England and account for
75% of the vegetated area in the Plum Island Sound estuary (Millette et al., 2010; Wilson et al., 2014). The high marsh exhibits
poor water drainage (Wilson et al., 2014) and is generally inundated biweekly during spring tides and during major storms (Millette
et al., 2010). The two dominant species on the high marsh are C4 species including *Spartina patens (*common name: marsh hay)
and *S. alterniflora* (common name: smooth cordgrass), with the latter mainly distributed along tidal channels and also dominant in
low marsh platforms (Anjum et al., 2012; Cheng et al., 2006; Curtis et al., 1990; Maricle et al., 2009; Sun et al., 2020). Another
C4 species, *Distichlis spicata* (coastal saltgrass), is often collocated within *S. patens*-dominated sites on high marsh platforms (Arp
et al., 1993), whereas *Juncus gerardii* (saltmarsh rush) usually dominates the terrestrial boundary of the high marsh (Bertness,

83    1991).

**2.2. Vegetation and Soil Sampling and Processing**
Aboveground biomass of the dominant species, *S. alterniflora* and *S. patens*, were collected every four to five weeks between June
and November in 2021, corresponding to the active growing season. Additional senesced biomass was sampled in the following
year in April 2022, and two additional salt marsh species, *D. spicata* and *J. gerardii,* were sampled in September 2018. For each
sampling date, eight 1-m$^2$ square plots were selected in the footprint area of a micrometeorological flux tower (Forbrich et al.,
2018), of which four squares were dominated by *S. alterniflora* and four adjacent squares were dominated by *S. patens*. During
vegetation sampling, all aboveground vegetation within the 1-m$^2$ squares was clipped close to the ground and stored in plastic
Ziploc bags in coolers over ice and subsequently in refrigerators until processing. In the laboratory, wet and dry vegetation mass
was determined, vegetation was carefully separated into live and senesced tissues and prepared for analysis of Hg in both bulk
samples and in individual species.
In four of the eight sampling sites, quantitative belowground sampling was performed in July 2021, with two plots dominated by
*S. alterniflora* and the other two plots dominated by *S. patens*. Soil cores with diameter of 10 cm to a depth of 40 cm were taken
and separated into depth increments of 0-20 cm and 20-40 cm. Belowground components were separated into the following
components by washing onto a fine mesh with pore size of 0.25 mm: live roots and rhizomes identified by turgidity and color (e.g.,
hard and white tissues versus soft and grey/discolored); senesced roots, rhizomes, and soil detritus (not recognizable organic matter);
and sediments and fine organic matter that passed through the fine mesh (only analyzed in two subsamples). All plant tissues were
rinsed with tap water until the water was clean, then thoroughly rinsed three times with Milli-Q water, while a selected number of
live aboveground tissues were analyzed both washed and unwashed for estimation of washable Hg (see section of throughfall
estimation). All samples were dried at 65 °C for at least 76 hours until constant weight, and ground using stainless steel coffee
grinders prior to analyses.
**2.3. Hg concentration and stable isotope analysis of vegetation and soils samples**
Total Hg concentrations in all components were measured using a tri-cell Milestone DMA-80 Direct Mercury Analyzer (Milestone
Inc., Monroe, Connecticut, USA) through thermal decomposition, catalytic reduction, amalgamation, desorption, and atomic
absorption spectroscopy following EPA method 7473 (U.S. EPA., 1998). The system was re-calibrated based on daily performance
checks using five-point calibration curves. Standard reference materials, including NIST 1515 Apple leaves (43.2 µg kg$^{-1}$) and
Canadian National Research Council certified reference material MESS-4 (marine sediment, 91 µg kg$^{-1}$), were used as continuous
calibration verifications after every ten-samples. Percent recoveries of total Hg for certified reference materials averaged of 99.9





± 5.5% (range of 89.6% to 111.4%) and all blanks were below detection limits (0.001 ng). All samples were analyzed in triplicate
and results were accepted when coefficients of variation were less than 10%.
Hg stable isotopes were measured on select samples including aboveground biomass, live root, live rhizomes, and surface (0-22.5
cm) and deeper soils (97.5 cm). Samples were pre-concentrated with a Nippon direct Hg analyzer (Nippon Instruments) as
described in Enrico et al. (2021). A HGX-200 cold vapor generator (Teledyne Cetac Technologies) was used to introduce sample
Hg to a Thermo Neptune plus MC-ICP-MS at Harvard University. An Apex-Q nebulizer (Elemental Scientific) was used to
nebulize a Thallium (Tl) solution and inject TI aerosols in the HGX-200. NIST3133 (primary standard) and RM8610 (previously
UM- Almaden, secondary standard) were used as Hg isotopic standard solutions, and NIST997 (thallium isotopic standard solution)
was used as the reference material to correct instrument mass bias. NIST 1515 Apple leaves and Canadian National Research
Council certified reference material MESS-4 were used to verify isotope analysis, and standard recoveries were in the acceptable
range (from 82% to 93%). Small delta ($\delta$) annotation is used for mass-dependent fractionation (MDF), which is reported as per mil
(‰) values relative to NIST-3133 based on equation (1),
$$\delta^{xxx}Hg = \left(\frac{(^{xxx}Hg/^{198}Hg)_{sample}}{(^{xxx}Hg/^{198}Hg)_{NIST3133}} - 1\right) \times 1000 \tag{1}$$
where $^{xxx}Hg$ is the mass of each Hg isotope between 199 and 204. Capital delta ($\Delta$) annotation is used for mass-independent
fractionation (MIF), describing fractionation away from the expected MDF based on equation (2),
$$\Delta^{xxx}Hg = \delta^{xxx}Hg - \beta_{xxx} \times \delta^{202}Hg \tag{2}$$
where $^{xxx}Hg$ denotes mass of each Hg isotope 199, 200, 201, and 204, and $\beta_{xxx}$ is the constant mass-dependent correction factor
(0.252, 0.502, 0.752, and 1.492, respectively; Blum and Bergquist, 2007). To determine Hg sources, a ternary isotope mixing
model was used to estimate fractions of Hg in above-ground biomass. End-member Hg sources used included signatures of salt
marsh plants roots, atmospheric GEM, and precipitation (see SI for details).
**2.4 Data Analysis**
Data were checked for normality (Shapiro–Wilk test) and homogeneity of variance assumptions of statistical tests. The non-
normalized data were subjected to a natural logarithmic transformation to ensure a normal distribution. Unpaired Student t-tests
were used to assess significant differences between groups (e.g., species), and statistical differences between non-washed and
washed aboveground vegetation samples were performed using paired Student t-tests. Linear regression analyses were performed
to determine the rate of aboveground biomass Hg uptake over time. Hg mass and turnover rates were calculated by multiplication
of Hg concentrations by corresponding biomass or biomass growth and other mass components at the level of sampling plots. All
statistical tests were performed with STATA (Version 16.0, Statacorps, College Station, Texas), and all regressions and statistical
tests presented in text, tables, and figures were based on statistical differences with p-values < 0.05. Variability presented in the
text and figures are standard deviations of means.
**3. Results**
**3.1 Hg concentrations in aboveground and belowground biomass**
Hg concentrations in aboveground tissues showed substantial seasonal variations and species-specific differences, with lowest
concentrations in live tissues of *S. alterniflora* and *D. spicata*, followed by *S. patens,* and highest concentrations in *J. gerardii* (Fig
1a). Despite species differences in Hg concentrations, concentrations in bulk vegetation of communities dominated by *S.*



*alterniflora* versus *S. patens* (Fig 1b) were not statistically different. This likely occurred because these communities are composed
of multiple species. For example, *S. alterniflora* communities also have a presence of *S. patens* plants*,* and *S. patens* communities
include large numbers of *D. spicata* plants. Similarly, Hg concentrations of senesced *S. patens* and *S. alterniflora* bulk samples
were not statistically significantly different from each other (Fig S1).
Hg concentrations in aboveground live biomass strongly increased throughout the growing season between June and November
across all species. Figure 2 shows a linear increase of Hg concentrations in live aboveground tissues in plots dominated by *S.*
*alterniflora* and *S. patens* over time ($r^2 = 0.84$; $p < 0.01$; $n = 50$), with no significant difference in regressions between the two
communities. Based on these linear regression slopes, we calculated daily uptake rates of Hg during the growing season of
$0.08\pm0.01$ µg kg$^{-1}$ day$^{-1}$ for both *Spartina* communities. After senescence, Hg concentrations in senesced aboveground biomass
measured in spring of the following year (April 2022) were not further enhanced compared to live biomass samples collected in
fall (November 2021; $p = 0.19$), (Figs 2 and 4a) so that no statistically significant Hg uptake (or loss) occurred in biomass after
senescence.
We washed and separated belowground samples into the following categories: live roots, live rhizomes, combined dead roots,
rhizomes, and detritus (unrecognizable biomass components), and combined fine soil mineral and humus fraction. This process
was based on visual separation of tissues (Elsey-Quirk et al., 2011; Valiela et al., 1976). We observed live roots and rhizomes only
in the top 20 cm of the soils with no recognizable live roots and rhizomes at 20-40 cm depth. Although aboveground Hg
concentrations between the two communities were similar, Hg concentrations in live roots and rhizomes (upper 20 cm)s were two
to three times higher in *S. patens* plots ($258.9\pm70.3$ µg kg$^{-1}$ and $46.6\pm14.2$ µg kg$^{-1}$ respectively) compared to *S. alterniflora* plots
($84.5\pm47.0$ µg kg$^{-1}$ and $27.9\pm1.1$ µg kg$^{-1}$ respectively) (Fig 3, Table S1). Compared to live tissues, we observed higher Hg
concentrations in senesced roots, rhizomes, and detritus ($318.0\pm30.1$ µg kg$^{-1}$ in *S. alterniflora*, $323.3\pm135.4$ µg kg$^{-1}$ in *S. patens*),
which also was higher than Hg concentrations in soil mineral and humus fractions of $272.3\pm11.6$ µg kg$^{-1}$ (although only measured
in one *S. patens* sample) (Fig 3, Table S1). Hg concentrations in senesced belowground biomass (roots, rhizomes, and detritus)
were higher than that in mineral and humus samples at 20-40 cm soil depths, although with larger variation (Table S1). Bulk soil
Hg concentrations (i.e., composed of all fractions listed above) averaged $194.6\pm28.3$ µg kg$^{-1}$ of *S. alterniflora* community and
$171.2\pm72.1$ µg kg$^{-1}$ of *S. patens* community in the top 20 cm with no significant difference ($p>0.05$). Bulk soil Hg concentrations
of the 20-40 cm soil in *S. alterniflora* ($279.1\pm203.8$ µg kg$^{-1}$) were almost twice that of *S. patens* ($159.1\pm122.7$ µg kg$^{-1}$). Overall,
Hg concentrations of live roots ($171.7\pm111.9$ µg kg$^{-1}$) were 11 times higher and live rhizome ($37.3\pm13.6$ µg kg$^{-1}$) were double the
concentrations of aboveground live biomass ($16.2\pm2.0$ µg kg$^{-1}$, Table S2).
**3.2 Hg pools sizes associated with aboveground and belowground biomass**
Aboveground standing live biomass strongly increased from June through August, when it plateaued at a peak biomass in August
($507\pm208$ g m$^{-2}$) and September ($498\pm118$ g m$^{-2}$, a trend that was consistent among species) (Fig 4b). Hg mass contained in live
aboveground biomass peaked later (in November) than standing biomass and showed an eight-fold and near-linear increase
between June ($0.7\pm0.4$ µg m$^{-2}$) and November ($5.7\pm2.1$ µg m$^{-2}$) (Fig 4c). Peak Hg pools contained in aboveground biomass were
$5.7\pm2.1$ µg m$^{-2}$ for live tissue and $3.3\pm1.7$ µg m$^{-2}$ for senesced tissue, for a total combined standing aboveground biomass Hg pools
of $9.0\pm3.3$ µg m$^{-2}$ in November (Figs 4c and 5). This also represents our best estimate of total annual Hg assimilation by
aboveground vegetation, assuming that little standing senesced biomass in November was attributable to NPP of the previous year
growing season. Standing aboveground biomass in the spring of the following year (April 2022, $357\pm148$ g m$^{-2}$) was 39% lower
than aboveground biomass in November of 2021 ($583\pm208$ g m$^{-2}$) (Fig 4b), and standing Hg pools were 32% lower in the





subsequent spring (6.1 ±1.9 µg m⁻²) compared to peak fall levels (9.0 ±3.3 µg m⁻²) (Fig 4c), showing losses of standing aboveground
biomass and associated Hg pools over winter.
Live root biomass in surface soils (top 20 cm) averaged 361±114 g m⁻² and live rhizome biomass were approximately twice as
large (792±231 g m⁻²), for a combined live belowground biomass of 1,153±321 g m⁻² (Table S2). Belowground Hg pools associated
with these live tissues averaged 70.0±63.7 µg m⁻² for roots, 38.1±22.4 µg m⁻² for rhizomes, and 108.1±83.4 µg m⁻² for the combined
live belowground tissue, accounting for less than 0.5% of the total bulk soil Hg pool (Fig 5a, b, Table S2). We observed a much
larger Hg pool associated with senesced biomass (roots, rhizomes, and detritus) averaging 4,116±1,141 µg m⁻², accounting for
16.1% of the total bulk soil Hg pool. We estimated a total soil Hg pool in the top 40 cm using measured bulk densities (range of
0.22 and 0.37 g cm⁻³) exceeding 25,000 µg m⁻², with most of this Hg associated with fine soil mineral and humus fraction (83.5%),
rather than contained in live and senesced plant tissues.
**3.3 Hg stable isotope signatures to determine Hg sources**
Aboveground biomass showed negative mass-dependent fractionation (MDF) values for $\delta^{202}$Hg between -1.61‰ and -1.07‰, and
mass-independent (MIF) values were consistently positive with $\Delta^{199}$Hg between 0.20‰ and 0.43‰ and $\Delta^{200}$Hg values between
0.04‰ and 0.11‰ (Fig 6, Tables S3 and S4). These aboveground isotopic Hg signatures of salt marsh vegetation fell outside of
the range commonly reported in foliar samples of terrestrial vegetation, both regarding mass-dependent and mass-independent
signatures. Specifically, terrestrial vegetation Hg signatures are substantially more negative in $\delta^{202}$Hg values (ranging from -3.06‰
to -2.37‰ [inter-quartile range, IQR, n = 120]) and both $\Delta^{199}$Hg and $\Delta^{200}$Hg values in terrestrial vegetation generally show negative
values ($\Delta^{199}$Hg: -0.42‰ to -0.27‰ IQR, $\Delta^{200}$Hg: -0.05‰ to 0.01‰, IQR) (Fig 6, Table S4) (review by Zhou et al., 2021).
Stable Hg isotope signatures of salt marsh plant roots were different from aboveground biomass, with less negative values for
$\delta^{202}$Hg (-0.75‰ and -0.66‰), less positive values for $\Delta^{199}$Hg (0.11‰ and 0.22‰), and close to zero values (instead of positive
values) for $\Delta^{200}$Hg (-0.01‰ and 0.04‰) (Fig 6, Tables S3 and S4). The Hg isotope signatures of roots closely overlapped with
signatures in surface marsh soils and deeper marsh soils ($\delta^{202}$Hg: -0.92‰ to -0.29‰, $\Delta^{199}$Hg: -0.09‰ to 0.20‰, and $\Delta^{200}$Hg: -
0.02‰ to 0.05‰, Tables S3 and S4). Similar to aboveground tissues, salt marsh soil isotopic Hg signatures were largely outside
the ranges reported for upland soils, particularly for $\delta^{202}$Hg values that are much more negative in upland soils ($\delta^{202}$Hg generally
between -0.5‰ and -2.9‰; review by Zhou et al., 2021). Hg isotope signatures of salt marsh rhizomes were quite variable and in-
between values observed in foliage and soils. Specifically, rhizomes showed $\delta^{202}$Hg values between -1.41‰ to -0.70‰, $\Delta^{199}$Hg
values between 0.13‰ to 0.22‰, and $\Delta^{200}$Hg values between -0.05‰ to 0.04‰ (Fig 6, Tables S3 and S4). Finally, $\Delta^{201}$Hg and
$\Delta^{199}$Hg across all marsh samples showed statistically significant correlations with a slope close to 1 (0.98, p < 0.01, Fig S2).
**4. Discussion**
**4.1 Salt marsh vegetation and soil Hg concentrations**
Strong seasonal Hg concentration increases in salt marsh aboveground tissues were consistent with patterns also observed in upland
ecosystems, such as in forest foliage (Wohlgemuth et al., 2022). In upland systems, foliar Hg increases are attributed in large part
to atmospheric GEM uptake, which is taken up during the growing season by stomatal and non-stomatal (i.e., cuticular) leaf uptake
(review by Zhou et al., 2021). Hg uptake is controlled by leaf physiological processes and related to photosynthetic capacity, leaf
nitrogen concentrations, leaf mass area, and stomatal densities and conductance (Wohlgemuth et al., 2022). In support of a similar
active role of plant physiology in controlling Hg uptake in salt marsh plants, we observed that Hg concentrations in senesced
biomass in April of 2022 were not significantly enhanced compared to live biomass of the previous November (2021) (Fig 2),





indicating that no significant Hg assimilation occurred during wintertime in senesced biomass. However, increases in Hg
concentrations occurred through November even after peak biomass was reached in August and September. We attribute this to
continued active plant physiology through late season similar to carbon assimilation and active plant photosynthesis which
continues at least through October at this site (Forbrich et al., 2018; no data is available for November). In contrast to upland plant
leaves, however, stable Hg isotope signatures of marsh aboveground biomass show different Hg sources and indicate Hg uptake
was not dominated by atmospheric GEM uptake (see below).
Calculated daily Hg accumulation rates in *Spartina*-dominated aboveground biomass (0.08 µg kg$^{-1}$ day$^{-1}$, Fig 2) was at the lower
range of foliar accumulation rates reported in forest foliage (conifer needle: median of 0.07 µg kg$^{-1}$ day$^{-1}$, deciduous leaf: median
of 0.23 µg kg$^{-1}$ day$^{-1}$; Wohlgemuth et al., 2022). This is consistent with the notion that low-statured grassland plants generally
exhibit lower Hg concentrations (5 µg kg$^{-1}$ [1-31 µg kg$^{-1}$]) than trees (e.g., forest foliage (20 µg kg$^{-1}$ [2-62 µg kg$^{-1}$]) (review by
Zhou et al., 2021), although it may also be due to different origins of Hg (section 4.2.1 below). Both dominant marsh species in
our study are C4 plants, which previous work shows have lower Hg concentrations compared to C3 species (e.g., 23±9 µg kg$^{-1}$
versus 53±12 µg kg$^{-1}$, Canário et al., 2017). In laboratory studies with upland plants, leaf uptake of Hg vapor has been linked to
catalase activity which is known to be lower in C4 plants (Du and Fang, 1983).
Highest Hg concentrations in *S. alterniflora* and *S. patens* were observed in fall (11.7 µg kg$^{-1}$ and 24.0 µg kg$^{-1}$, respectively). Hg
concentrations fell within concentration ranges reported from other uncontaminated marsh halophytes (Table S6), such as 5 to 33
µg kg$^{-1}$ in the Great Bay estuary in New Hampshire, USA (Heller and Weber, 1998), an average of 20 µg kg$^{-1}$ in Big Sheepshead
Creek estuary in New Jersey USA (Kraus et al., 1986), and 3 to 79 µg kg$^{-1}$ in the Ria de Aveiro Coastal Lagoon, Portugal (Anjum
et al., 2011). Hg concentrations in aboveground tissues in our study, however, were much lower than those from contaminated
marshes where concentrations up to 90 µg kg$^{-1}$ were observed in the Hackensack Meadowlands in New Jersey (Windham et al.,
2001) and up to 160±70 µg kg$^{-1}$ in Piles Creek in New Jersey (Kraus et al., 1986) and even up to 1124 ± 21 µg kg$^{-1}$ in Tagus
estuary, Portugal (Canário et al., 2017). Lower Hg concentrations (10.2 ± 0.9 µg kg$^{-1}$) were reported from the polluted Yangtze
River estuary (Wang et al., 2021).
In contrast to upland plants, salt marsh plants (including both *Spartina* species) have salt glands which are used for selective and
active excretion of sea salt (Kirschner and Zinnert, 2020; Maricle et al., 2009). Salt glands also have been linked to excretion of
metals (Weis and Weis, 2004), and previous studies reported correlations between leaf surface Hg and sodium (Na) release in *S.*
*alterniflora* suggesting active Hg excretion by salt glands (Weis and Weis, 2004; Windham et al., 2001). Windham et al. (2001)
proposed that in the Hackensack Meadowlands, a polluted salt marsh ecosystem, seasonal declines in Hg concentrations in *S.*
*alternifla* leaves between May (90 µg kg$^{-1}$) to July (30 µg kg$^{-1}$) were driven by strong leaf excretion of Hg. By washing leaves of
a select number of samples, we found that washing removed about 6% of total leaf Hg in *S. alterniflora* and 16% in *S. patens*,
respectively (Table S5, note that we use the wash-off fraction as an estimate of throughfall deposition in the mass balance
estimation below). The relatively minor loss of Hg associated with washing showed that most leaf Hg was structural and likely
internal Hg, which along with observed seasonal Hg concentration increases does not support substantial seasonal Hg losses nor
seasonal concentration declines which would be attributable to salt excretion at this site.
Hg concentrations in live roots and rhizomes at our sites were much higher (11 and two times, respectively) than aboveground live
biomass concentrations. This is consistent with previously reported data that also reported higher root Hg concentrations in salt
marshes (Anjum et al., 2012; Cabrita et al., 2019; Canário et al., 2017; Garcia-Ordiales et al., 2020; Weis and Weis, 2004; Windham
et al., 2003). Hg concentrations were higher in live roots of *S. patens* (258.9±70.3 µg kg$^{-1}$) compared to *S. alterniflora* (84.5±47.0
µg kg$^{-1}$) (Fig 3, Table S1). A possible reason for this is finer roots in *S. patens* (personal observation), and hence higher surface to
volume ratios, which may facilitate soil Hg uptake. In upland ecosystems, fine root Hg concentrations were reported to be higher



than in coarse roots as well (Wang et al., 2012; Wang et al., 2020). Elevated root and rhizome Hg concentrations compared to
aboveground tissues in marsh plants contrast with upland studies that generally report higher concentrations in foliage (20 µg kg
$kg^{-1}$ [2–62 µg $kg^{-1}$]) and much lower concentrations in roots (7 µg $kg^{-1}$ [2–70 µg $kg^{-1}$]) (Zhou et al., 2021). An exception to this are
grassland systems where root Hg concentrations also have been reported higher than foliage, although the difference was much
smaller (e.g., roots: 41±31 µg $kg^{-1}$; leaves: 20±10 µg $kg^{-1}$; (Zhou and Obrist, 2021)). Our data also showed much lower root Hg
concentrations in this marsh compared to a moderately contaminated estuary in Portugal (Anjum et al., 2012; Canário et al., 2017;
Garcia-Ordiales et al., 2020), with the exception of lower root Hg in a contaminated salt marsh roots in the Yangtze River estuary
(Wang et al., 2021) (Table S6). Overall, the reported results suggest a large range of Hg concentrations in belowground salt marsh
biomass, which likely is dependent on soil Hg concentrations as the dominant Hg source for root Hg (see section 4.2.1 below).
Much higher concentrations of root and rhizomes in belowground tissues compared to aboveground biomass also suggests limited
translocation between belowground and aboveground tissues (Cavallini et al., 1999; Clemens and Ma, 2016; Graydon et al., 2009;
Wang et al., 2012) and possibly different sources between the two (as discussed below).
**4.2 Stable Hg isotope signatures and possible origins of Hg in salt marsh vegetation and soil**
**4.2.1 Salt marsh vegetation**
One of the largest MDF processes in the environmental systems is due to preferential uptake of light atmospheric GEM isotopes
by vegetation foliage that leads to large negative $\delta^{202}Hg$ signatures (generally below -2‰) and mass independent signatures similar
to that of atmospheric GEM (Demers et al., 2013; Enrico et al., 2016; Yu et al., 2016). In terrestrial ecosystems, studies have shown
that the vegetation uptake of atmospheric GEM and subsequent litterfall, throughfall, and plant senescence serves as the primary
source of Hg loading (Demers et al., 2013; Jiskra et al., 2015; Louis et al., 2001; Obrist et al., 2017; Wang et al., 2016; Zheng et
al., 2016; Zhou et al., 2021). Aboveground biomass of salt marsh plants show a distinctly different signature than reported patterns
from upland foliage. Specifically, MDF values were much less negative, and values of odd-MIF ($\Delta^{199}Hg$) and even-MIF ($\Delta^{200}Hg$)
were more positive, compared to upland foliage (Fig 6, Table S4). We propose that salt marsh plant leaves have distinctly different
sources than the dominant atmospheric GEM uptake proposed in upland plants. The similarities of the two MIF patterns ($\Delta^{199}Hg$
and $\Delta^{200}Hg$) further suggest that the difference is largely due to different sources (i.e. end-member mixing) as opposed to process-
based fractionation processes after uptake (review by Kwon et al., 2020).
Hg signatures of salt marsh aboveground tissue were close to signatures of salt marsh soils, yet with slightly more negative $\delta^{202}Hg$
values and more positive $\Delta^{199}Hg$ and $\Delta^{200}Hg$ values (Table S4). We used a ternary mixing model to identify Hg sources and further
quantify their contributions for salt marsh plant leaves based on MDF ($\delta^{202}Hg$) and even-MIF ($\Delta^{200}Hg$) (Demers et al., 2013; Jiskra
et al., 2021; Jiskra et al., 2017; Obrist et al., 2017). Briefly, the dominant three end-member Hg sources include: (1) direct uptake
from marsh plant roots, (2) atmospheric GEM uptake through leaf stomata, and (3) precipitation Hg(II) deposition.
Our best estimate shows that the Hg source in salt marsh vegetation consists of a mixture of atmospheric GEM of 32%, root uptake
of 35%, and precipitation deposition of around 33%. Most notably, the biggest difference compared to upland plants is much less
negative $\delta^{202}Hg$ values. We propose that some uptake of atmospheric GEM leads to $\delta^{202}Hg$ values that are more negative than the
$\delta^{202}Hg$ of plant roots and soils. Precipitation, on the other hand, which largely consists of oxidized Hg, shows a typical positive
anomaly in $\Delta^{200}Hg$ linked to upper atmosphere GEM oxidation (Enrico et al., 2016; Jiskra et al., 2021; Zhou et al., 2021). We
propose that precipitation contributions caused a partial $\Delta^{200}Hg$ anomaly in salt marsh aboveground biomass compared to soil
sources. Our results suggest a more important role of Hg transport from belowground (i.e., roots) to aboveground tissues in salt
marsh vegetation compared to upland ecosystems that report minor translocation of Hg from belowground to aboveground tissues
(generally below 5% of leaf Hg originating from soils via root uptake, review by Zhou et al., 2021). Vegetation studies from salt





marshes previously suggested inconsistent leaf Hg source patterns. For example, an Hg isotope tracer study suggested minor root-
to-leaf transport with soils accounting for a small percentage of Hg in marsh plants (i.e. 2.2-2.7% from Cabrita et al. (2019)), while
a study based on bioaccumulation factors suggested a wide and inconstant range of soil Hg contribution to leaves from 1.7-9.6 %
to as high as 46% (Castro et al., 2009).
The mixing model results were not able to fully match the range of odd-MIF ($\Delta^{199}$Hg) due to more positive $\Delta^{199}$Hg signatures in
salt marsh plant leaves, and it is possible that chemical and biological processes modify original salt marsh vegetation isotope
signatures after uptake. For example, odd-MIF signature is impacted by photochemical reductions of aqueous inorganic Hg(II)
inducing more positive values (Bergquist and Blum, 2007; Kwon et al., 2014; Meng et al., 2019; Yuan et al., 2019). Relationships
between two odd-MIF, $\Delta^{201}$Hg and $\Delta^{199}$Hg, are used to assess photoreduction pathways (Bergquist and Blum, 2007), and we
observed a positive correlation between $\Delta^{201}$Hg and $\Delta^{199}$Hg across salt marsh samples (slope of 0.98; Fig S2), which is in large
parts driven by leaves with higher values of both $\Delta^{201}$Hg and $\Delta^{199}$Hg values. This slope is close to a slope reported during inorganic
Hg(II) photoreduction (slope of 1.00) (Bergquist and Blum, 2007; Blum et al., 2014), so that it is possible that photochemical
reduction of Hg in exposed leaves may contribute to isotopic patterns.
Roots of salt marsh plants show a Hg isotope signature that almost perfectly overlaps the signatures observed in soils, strongly
suggesting a dominant soil source. In terrestrial plants, Hg assimilated in belowground biomass also is considered largely of soil
origin with little internal translocations of Hg from aboveground tissues (Millhollen et al., 2006; Obrist et al., 2018; Zhou et al.,
2021). This also has been proposed in aquatic plants (e.g., mangroves, sawgrass) where root Hg largely derives from surrounding
soils (Huang et al., 2020; Mao et al., 2013; Yin et al., 2013). Dominant soil Hg sources in roots would also explain the differences
in root Hg levels with salt marsh contamination levels as discussed above. Finally, rhizome Hg isotope signatures indicate a mix
of above-ground and belowground Hg sources, although they show a large variation in isotope signatures with some samples being
closer to aboveground tissue and others being closer to root signatures. This observation is consistent with the role of rhizomes as
storage organs with over one year lifetime, whereby carbohydrates and nutrients are mobilized via rhizomes between above- and
belowground organs based on plant allocation needs.
**4.2.2 Salt marsh soil**
The salt marsh soil isotope signature fell almost completely outside the range of soil Hg signatures reported from upland studies.
In terrestrial environments, the strong MDF during foliar GEM uptake imprints a similar and typical terrestrial fingerprint on soil
Hg, resulting in soil signatures with strong negative $\delta^{202}$Hg and $\Delta^{200}$Hg values similar to that of vegetation. Mixing models based
on isotopic $\Delta^{200}$Hg data suggest upland soil Hg sources are dominated by atmospheric GEM (accounting for 53% to 92% of the
source), which originates from plant Hg uptake and subsequent deposition (e.g., plant senescence) of overlying vegetation (Jiskra
et al., 2018; Obrist et al., 2018; Zhou et al., 2021; Zhou and Obrist, 2021). These upland soil Hg isotope signatures propagate in
watershed runoff (Jiskra, et al., 2017; Woerndle et al., 2018). Soils of our salt marsh study notably lacked the strong $\delta^{202}$Hg
depletion signal of uplands soils (e.g., $\delta^{202}$Hg of marsh soils between -0.92‰ and -0.29‰, versus -0.5‰ to -2.5‰ in other soils,
review by Zhou et al., 2021) and further supports that the source of Hg in marsh vegetation, which ultimately deposits to soils, is
distinctly different from that of upland ecosystems. The isotopic signature of soil samples in Figure 6a also does not support a
simple two-way mixing between plant and precipitation Hg to explain salt marsh soils Hg signatures. Further terrestrial surface
runoff, which generally shows typical terrestrial origin signatures (but was not measured in our study), also cannot explain marsh
soil Hg isotope patterns.
Hg isotopic signatures of this marsh soils strongly overlap within both ocean signatures as well as with reported industrial and
legacy contamination that are additional potential Hg sources in salt marsh soils. Mixing models, however, cannot be used to



calculate their potential contributions due to very large variations of these end-member Hg sources that overlap with the signatures
in soils. Seawater regularly floods the salt marsh during spring tides and storms and provide solids for salt marsh soils (Millette et
al., 2010). Recently reported ocean water Hg isotopes show total Hg median values for $\Delta^{200}$Hg of 0.02‰ (-0.01‰ to 0.03‰ IQR),
while ocean particulate Hg showed similar (albeit more variable) patterns (Jiskra et al., 2021). Adding salt marsh vegetation Hg
isotope signatures to those of ocean water reported by, we observe a strong overlap of salt marsh soil with ocean signatures,
whereby both $\Delta^{200}$Hg and $\delta^{202}$Hg fall between the ranges reported for seawater (Fig S3, note that due to large variability, we were
unable to quantify respective source fractions). The notion of ocean Hg sources would be consistent with a sediment mass balance
study which showed that sediment loads were dominated by ocean sediments in the Plum Island Sound estuary salt marsh
(Hopkinson et al., 2018), with relatively minor import of sediments derived from the watershed. Industrial and legacy
contamination sources also may shape salt marsh soil Hg signatures. Industrial Hg isotope signatures are characterized by large
ranges of negative $\delta^{202}$Hg values and near-zero to positive $\Delta^{199}$Hg and $\Delta^{200}$Hg values (Fig S3, Table S4, see SI for details).

**4.3 Hg mass balance and turnover fluxes associated with biomass dynamics.**

**4.3.1 Aboveground**

Aboveground biomass turnover normally dominates atmospheric Hg deposition in terrestrial systems, For example, across 16 states
in the eastern U.S., median annual aboveground litterfall Hg deposition, ultimately deriving from GEM uptake, was 11.7 µg m$^{-2}$
yr$^{-1}$ (range of 2.2-23.4 µg m$^{-2}$ yr$^{-1}$) and exceeded annual wet Hg deposition by rain (median of 9.2 µg m$^{-2}$ yr$^{-1}$; range of 4.5-19.7 µg
m$^{-2}$ yr$^{-1}$) (Risch et al., 2017). The implication of aboveground tissue turnover for Hg cycling in salt marshes are likely distinctly
different from terrestrial systems due to different Hg sources. Estimated annual aboveground Hg assimilation by salt marsh plants
is 9.0±3.3 µg m$^{-2}$ yr$^{-1}$. Of this turnover, however, our stable isotope data suggest that about 65% (i.e., 5.9±2.1 µg m$^{-2}$ yr$^{-1}$) constitutes
an external source from atmospheric GEM uptake and from precipitation, and the rest (35%, 3.1±1.1 µg m$^{-2}$ yr$^{-1}$) likely originates
from soil uptake and hence represents an internal plant-soil recycling of Hg within the ecosystem.
Based on these results, we estimate here a mass balance of Hg sources and sinks associated with aboveground vegetation dynamics
and turnover and compare these with previously reported fluxes such as lateral tidal exchanges, published wet and gaseous oxidized
Hg, and particulate Hg deposition. Hg inputs to this salt marsh include wet Hg deposition, which based on interpolated data by the
NADP program is estimated at 5.2 µg m$^{-2}$ yr$^{-1}$ (NADP, 2017), while a lower estimate of 2.9 µg m$^{-2}$ yr$^{-1}$ has been measured at a
nearby coastal site on Cape Cod, Massachusetts (Engle et al., 2010). Combining these two data sets, we estimate a mid-point wet
deposition of 4.1 µg m$^{-2}$ yr$^{-1}$. Gaseous oxidized Hg (GOM) and particulate Hg (PHg) deposition in this area was estimated at 1.2
µg m$^{-2}$ yr$^{-1}$ based on measurements by Engle et al. (2010) and at 3.0 µg m$^{-2}$ yr$^{-1}$ at a deciduous forest (Harvard Forest) in
Massachusetts (Obrist et al., 2021). Hence, a mid-point dry deposition of combined GOM and PHg is estimated at 2.1 µg m$^{-2}$ yr$^{-1}$
(Table 1, Fig 7). Aboveground vegetation Hg dynamics yields a total turnover of 9.0±3.3 µg m$^{-2}$ yr$^{-1}$ (combined live and senesced
biomass at the end of the growing season), including 5.9±2.1 µg m$^{-2}$ yr$^{-1}$ constitutes an atmospheric GEM source and root uptake
represents 3.1±1.1 µg m$^{-2}$ yr$^{-1}$. Based on sample washing, an additional 11% of foliar Hg concentrations was subject to wash-off
so that we constrain throughfall deposition of Hg to 1.0±0.4 µg m$^{-2}$ yr$^{-1}$. Combined atmospheric Hg deposition attributable to
aboveground vegetation hence yields 6.9 µg m$^{-2}$ yr$^{-1}$, and is closer to the combined wet, GOM, and PHg deposition (6.2 µg m$^{-2}$ yr$^{-1}$
$^{-1}$). Combined atmospheric Hg sources in this system are estimated at 13.1 µg m$^{-2}$ yr$^{-1}$ (Table 1; a range of 7.7 to 19.5 µg m$^{-2}$ yr$^{-1}$).
Aboveground vegetation also results in lateral exchange of Hg between marsh and tidal water via wrack export, i.e., losses of
plants and surface litter through tidal flushing. Although difficult to measure, wrack export in this area is composed primarily of
*S. alterniflora* plants (Hartman et al., 1983) and has been estimated to constitute 16-19% (mid-point of 17.5%) of biomass that
accumulates from NPP in the marsh (Duarte, 2017; Duarte and Cebrián, 1996). Hence, we estimate that of the annual Hg uptake



by aboveground biomass (of 9.0 ±3.3 µg m⁻² yr⁻¹) about 1.6 µg m⁻² yr⁻¹ (range of 1.4-1.7 µg Hg m⁻² yr⁻¹) may be subject to wrack
export. Scaling to the whole salt marsh area, the total wrack export from this marsh is estimated around 0.06 kg yr⁻¹ (range of 0.06-
0.07 kg yr⁻¹). In a previous study, we quantified Hg exports from the salt marsh system via tidal exchanges of dissolved and
particulate Hg (without plants), and estimated 0.7 µg m⁻² yr⁻¹ of dissolved Hg export and 5.6 µg m⁻² yr⁻¹ of particulate Hg export
from the marsh to the tidal water (Wang and Obrist, 2022). Our estimated annual Hg export by wrack is higher than lateral export
of dissolved Hg, but much smaller than lateral export of particulate Hg. Considering all these inputs and outputs, we estimate a net
present-day Hg mass accumulation in this salt marsh ecosystem between 0-11.5 µg m⁻² yr⁻¹ with a mid-point of 5.2 µg m⁻² yr⁻¹,
suggesting that this salt marsh currently represents a small net sink of environmental Hg.
It is noteworthy that the timing of aboveground tissue turnover in salt marshes occurred later than that of upland forests that have
litterfall Hg inputs mainly in the fall. We observed relatively slow losses of senesced biomass over winter, whereby standing
senesced aboveground biomass in spring of the subsequent year (April of 2022 357±148 g m⁻²) was only 39% lower than peak
aboveground biomass in November of 2021 (583±208 g m⁻², Fig 4b). Zawislanski et al. (2001) similarly reported that 32% to 39%
of leaf mass was still attached to stems seven months after senescence in *S. alterniflora* stands in May of the subsequent year
compared to the previous September. Senesced biomass finally is incorporated into soils, exported as wrack to the ocean, lost to
decomposition, or subject to herbivory. Zawislanski et al. (2001) summarized studies and discussed that a large part of NPP (60%
to 80%) accumulated in salt marsh soils; Duarte (2017) and Duarte and Cebrián (1996) suggested that the largest component of
NPP (43%) was decomposing in the system, 14-17% of NPP was subject to long-term burial, and smaller amounts were subject to
ocean export as wrack (16-19% of NPP) and consumed by herbivores (27%;). Another study estimated NPP loss due to herbivory
of 5% (Mann, 1988). Based on these studies, we estimate that of the annual Hg mass assimilated in aboveground biomass, the
largest fraction (57% to 80%, equivalent to 5.1-7.2 µg Hg m⁻² yr⁻¹) remained in the system and was subject to net accumulation
and decomposition, 16-19% (1.4-1.7 µg Hg m⁻² yr⁻¹) was subject to wrack export, and 5% to 27% (0.5-2.4 µg Hg m⁻² yr⁻¹) was
subject to herbivory.
**4.3.2 Belowground**
Many studies show that in salt marsh ecosystems, belowground productivity generally is equal or greater than aboveground
biomass production, and this particularly applies for northern marshes (Blum, 1993; Morris, 2007; Tobias and Neubauer, 2019;
Windham, 2001). Roots of both dominant species can grow to length of 8 to 20 cm (Blum, 1993; Muench and Elsey-Quirk, 2019).
*S. alterniflora* normally has large and thick rhizomes (normally ranging from 2-4 mm in diameter) with aerenchyma tissues to
transport oxygen to submerged belowground tissue for respiration, while *S. patens* has relatively dense and fine roots with limited
aerenchyma tissue which cannot support aerobic respiration when completely flooded (Muench and Elsey-Quirk, 2019). Live root
biomass (upper 40 cm) of 444±87 g m⁻² in *S. patens* and 278±61 g m⁻² in *S. alterniflora* cores (Table S1), is consistent with reported
denser root biomass in *S. patens* compared to *S. alterniflora* (Muench and Elsey-Quirk, 2019). Combined live roots and rhizome
biomass averaged 1,153±321 g m⁻², and thereby exceeded peak standing aboveground biomass of 830±415 g m⁻² in August 2021.
Scaling up Hg pools using these belowground biomass data and measured Hg concentrations yields large belowground Hg pools.
For example, the live belowground Hg pool (roots and rhizomes) is 108.1±83.4 µg m⁻² and more than ten times larger than peak
standing aboveground Hg pools (9.0±3.3 µg m⁻²) (Fig 5a, Table S2). The Hg pool associated with senesced biomass (roots,
rhizomes, and detritus) was over an order of magnitude larger (4,116±1,141 µg m⁻²).
Turnover times of salt marsh macrophyte roots are estimated at 0.6 yr⁻¹ (0.2 to 1.9 yr⁻¹ ) (Ouyang et al., 2017) and 0.5 yr⁻¹ (Blum,
1993), although longer turnover times have been proposed for creek-side plants (2.6 yr⁻¹ , Blum, 1993). Assuming a belowground
biomass turnover rate of 0.6 yr⁻¹ (0.2 to 2.6 yr⁻¹), estimated Hg mass turnover associated with belowground biomass (root and



rhizome) is 58.6 µg m$^{-2}$ yr$^{-1}$ (19.5 to 253.8 µg m$^{-2}$ yr$^{-1}$) (Table 1, Fig 7). Hence, belowground Hg turnover via plant tissues exceeds that of aboveground tissue (9.0±3.3 µg m$^{-2}$ yr$^{-1}$) by a factor five, although it is largely unclear what the implications of this turnover may be. Given that we consider the source of belowground tissue Hg to be largely from soil uptake, this large Hg belowground turnover flux does not provide an external source and represents internal recycling of Hg between soils and belowground tissues. This recycling of Hg may have been various consequences, such as impacting mobility and bioavailability, phytostabilization by roots (Anjum et al., 2011), or remobilization of Hg associated with root decomposition.

### 5. Summary and conclusion

Measurements of Hg concentrations, fluxes, and turnover associated with vegetation in a salt marsh ecosystem with high above- and belowground NPP showed an annual Hg uptake in aboveground tissues of 9.0 ±3.3 µg m$^{-2}$ yr$^{-1}$. Using a stable Hg isotope mixing model, we estimate that 35% of aboveground Hg originates from soil Hg uptake, 32% is from atmospheric GEM uptake, and 33% is from precipitation Hg(II) deposition. Estimated annual plant-derived atmospheric Hg deposition from plant senescence (i.e., litterfall) is estimated at 5.9±2.1 µg m$^{-2}$ yr$^{-1}$, which is about half of that in forests where plant Hg assimilation of atmospheric GEM is the dominant Hg source. We estimate an additional atmospheric Hg deposition by throughfall of 1.0±0.4 µg m$^{-2}$ yr$^{-1}$, for combined plant-derived Hg inputs of 6.9 µg m$^{-2}$ yr$^{-1}$. This deposition is similar to combined wet and dry deposition of other atmospheric Hg forms. Seasonal and temporal Hg concentration and mass balance dynamics show strong seasonal increases during active growing season and a lack of concentration changes after senescence over winter, suggesting physiologically controlled uptake pathways. Hg contained in aboveground tissues lead to an annual wrack export (losses to tidal flushing) of 1.6 µg m$^{-2}$ yr$^{-1}$ to tidal water and ocean and herbivory of Hg in a range of 0.5 to 2.4 µg Hg m$^{-2}$ yr$^{-1}$ (Table 1, Fig 7). The remainder of vegetation Hg is slowly incorporated into soils over winter and during the subsequent year.

Belowground Hg pools associated with live tissues collected in July (108.1±83.4 µg m$^{-2}$) were over ten times larger than peak aboveground Hg pools and resulted in a substantial annual Hg turnover flux of 58.6 µg m$^{-2}$ yr$^{-1}$. The source of root Hg is largely from soil uptake, while belowground rhizomes show variable sources both from aboveground and root tissues. The turnover of Hg associated with belowground tissues largely reflects internal recycling between soils and plants, with poorly understood impacts on Hg partitioning, bioavailability, and mobility. Hg associated with roots and rhizomes only accounted for about 0.4% of total belowground Hg pools, with the largest soil Hg pools associated with fine soil mineral and humus fractions (83.5%). Overall, we estimate this marsh to presently serve as a small net Hg sink for environmental Hg of 5.2 µg m$^{-2}$ yr$^{-1}$.

### 6. Competing interests

The contact author has declared that none of the authors has any competing interests.

### 7. Acknowledgements

We thank Keely O'Beirne, Madison Sachs, and Silas Bollen for help with vegetation sampling and laboratory analysis of initial plant and soil mercury samples. We thank Nancy Pau and the Parker River National Wildlife Refuge for sampling permits and acces. We thank Samuel Kelsey, Anne Giblin and other researchers from the Plum Island Ecosystem Long-Term Ecological Research project for support and information about the estuary. Funding was provided by an award from the U.S. National Science Foundation Division of Environmental Biology (Award Number: 2027038).



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





**Figures and tables**



Figure 1. Seasonal Hg concentrations of the four dominated salt marsh live plant species in 2018 and 2021 a), and seasonal Hg concentrations of the *S. alterniflora* and *S. patens* communities in 2021 b). Different colors indicate different plant species. Standard errors indicate four replicates. *: Standard errors indicate duplicates for a sample.





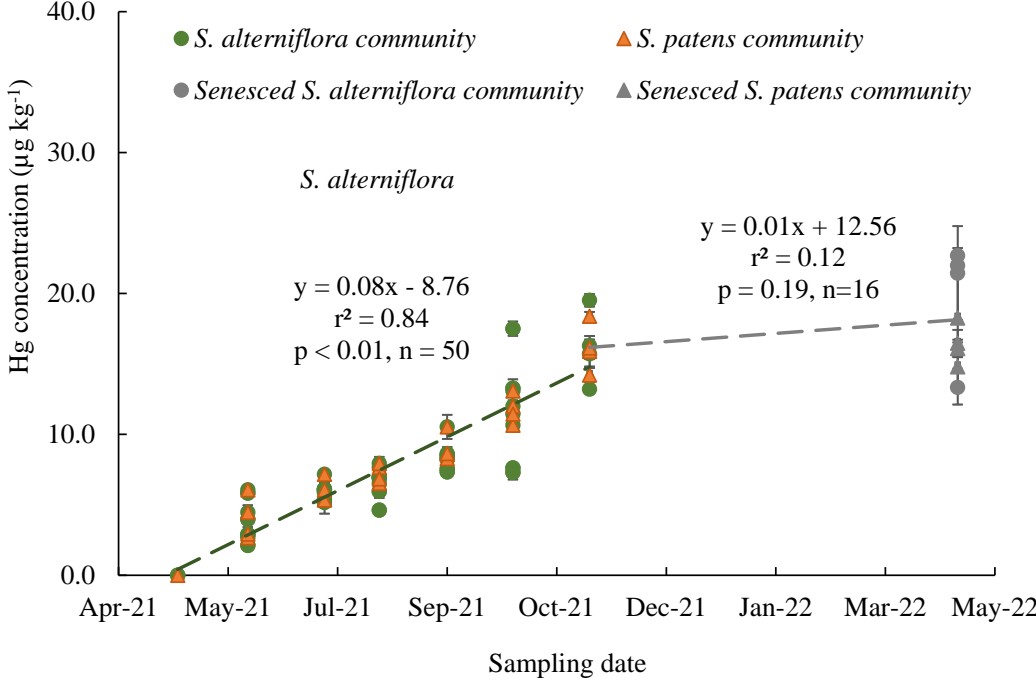

723

**Figure 2. Hg concentrations of live and senesced aboveground biomass of S. alterniflora and S. patens communities corresponds with sampling dates in 2021. Green circles indicate live S. alterniflora communities, orange triangles indicate live S. patens communities, grey circles indicate senesced S. alterniflora communities, and grey triangles indicate senesced S. patens communities. Standard errors indicate four replicates.**

728

729



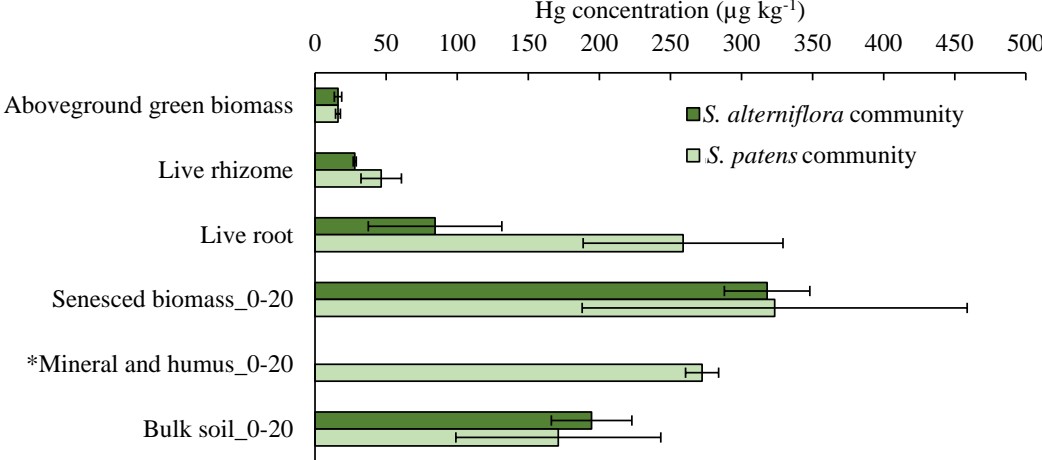

730

**Figure 3. Hg concentrations in live above- and belowground biomass of *S. alterniflora* and *S. patens* communities, as well as Hg concentrations of minerals and humus and bulk soils up to depth of 20cm covered by these two plant species. Dark green columns denote *S. alterniflora* community, light green columns denote *S. patens* community. Standard errors indicate multiple sample analysis. \*Hg concentration in mineral and humus only present one site covered by *S. patens*, and standard errors are duplicates.**



**Figure 4 Seasonal patterns of Hg concentrations a), biomass dry weight b), and Hg mass c) in aboveground live and senesced biomass from June 2021 to April 2022. The green columns represent live biomass, grey columns represent senesced biomass, and orange columns represent total biomass weight and Hg mass of adding live and senesced biomass. Standard errors indicate four replicates.**





Figure 5. Hg mass of above- and belowground biomass, including live and senesced biomass, and mineral and humus fractions in a soil depth of 40cm, a), and percentages of Hg mass contribution from belowground sections to the Hg soil pool of a soil depth of 40cm, b). Different colors indicate of different sections of the marsh.



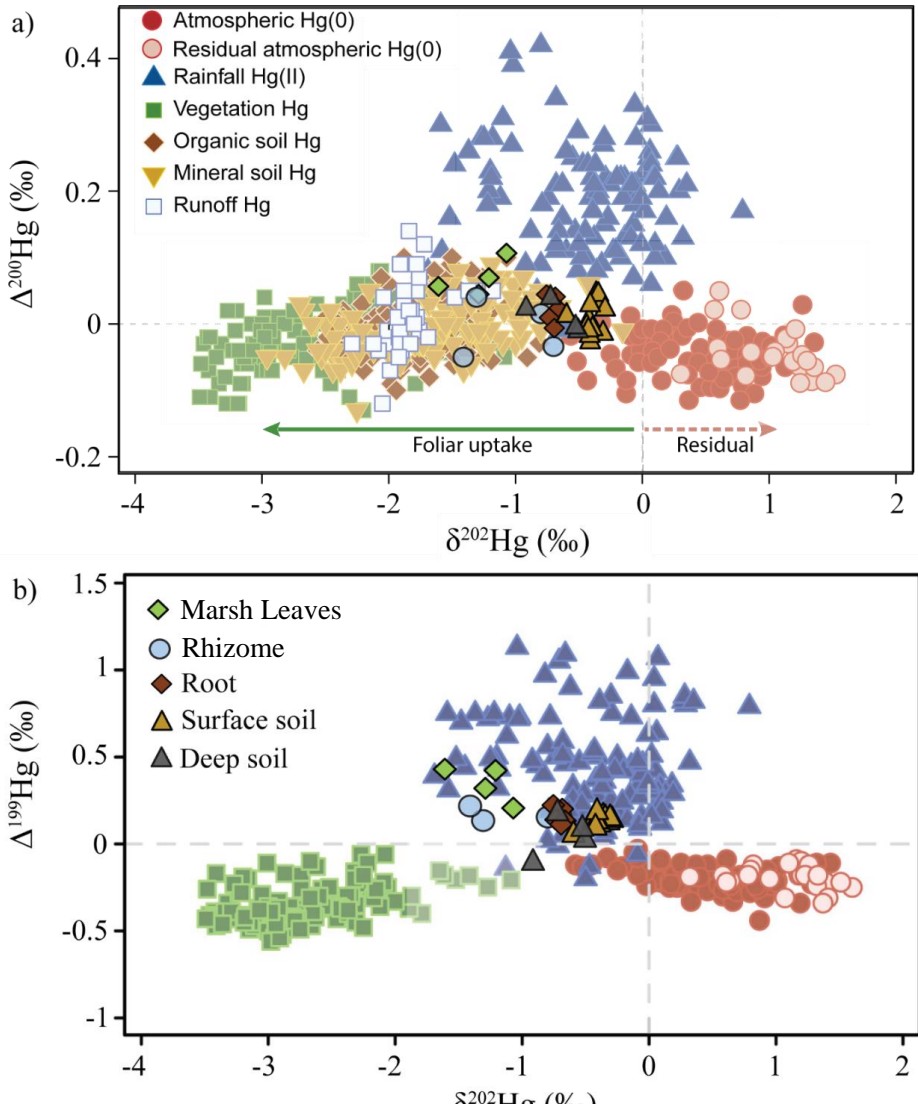

**Figure 6. Hg isotopes in salt marsh plants and soils, and foliage a) of $\Delta^{200}$Hg and $\delta^{202}$Hg, and b) of $\Delta^{199}$Hg and $\delta^{202}$Hg. Composition of Hg sources in marsh vegetation and soils (surface and deep soil layers), and all previously published currently available isotope data of sources of Hg in vegetation and in terrestrial sinks, atmospheric Hg(0) and Hg(II) sources (Zhou et al., 2021), plotted as a) even-mass-independent ($\Delta^{200}$Hg) versus mass-dependent ($\delta^{202}$Hg) isotopes a) and b) odd-mass-independent ($\Delta^{199}$Hg) versus mass-dependent ($\delta^{202}$Hg) isotopes.**




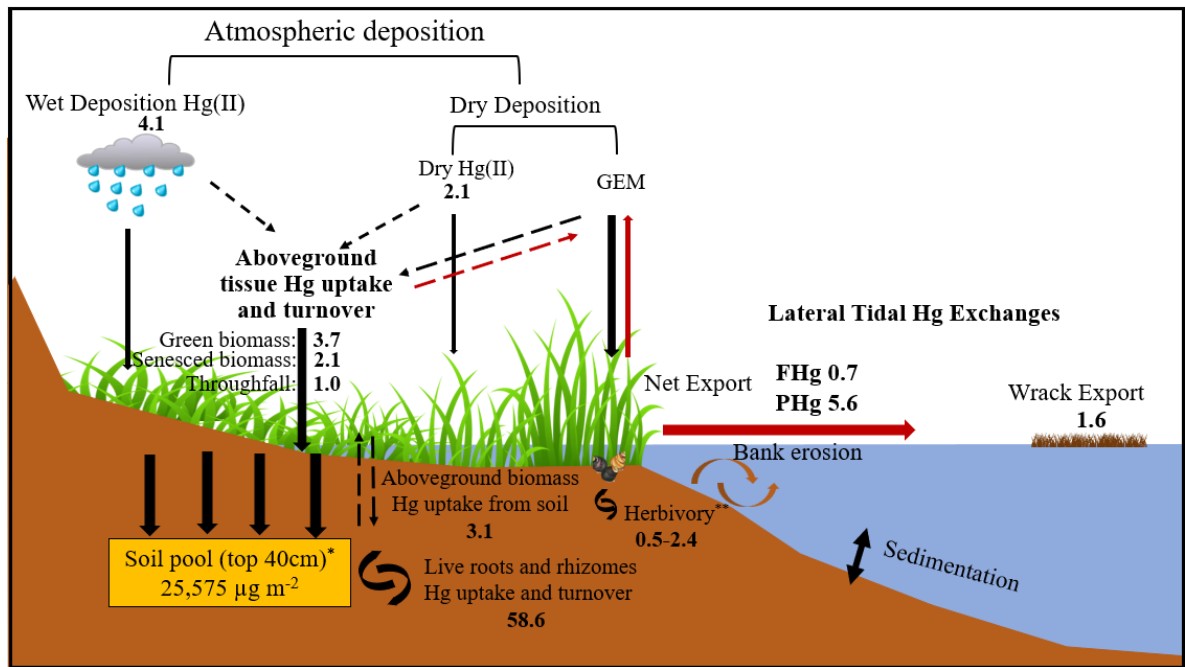


**Figure 7. Hg mass balance of the study salt marsh ecosystem. The values shown mostly represent the median Hg flux values with unit of μg m$^{-2}$ yr$^{-1}$, except for the soil pool$^*$, which represents an averaged value, and herbivory$^{**}$, which indicates a range. The red arrows indicate the emission of Hg back to the atmosphere and its export out of the salt marsh.**







**Table 1 Hg mass balance of the study salt marsh ecosystem.**

| | Category | Item | Hg flux (μg m⁻² yr⁻¹) | Percent of Hg sources | Reference | Hg fluxes scaled to the marsh (kg yr⁻¹)* |
|---|---|---|---|---|---|---|
| **Present-day Hg Mass Balance (total inputs minus exports)** | Deposition | Green aboveground biomass* | 3.7±1.4 (1.3-5.0) | 65% atmospheric Hg | This study | 0.15 (0.05-0.20) |
| | | Senesced aboveground biomass* | 2.1±1.1 (1.2-4.7) | 65% atmospheric Hg | This study | 0.08 (0.05-0.19) |
| | | Total aboveground biomass | 5.9±2.1 (3.1-9.7) | 65% atmospheric Hg | This study | 0.24(0.12-0.39) |
| | | Throughfall | 1.0±0.4 (0.5-1.6) | 100% atmospheric Hg | This study | 0.04(0.02-0.06) |
| | | Wet Hg(II) | 4.1 (2.9-5.2) | 100% atmospheric Hg | (Engle et al. 2010, NADP, 2017) | 0.16 (0.12- 0.21) |
| | | Dry Hg(II) | 2.1 (1.2-3.0) | 100% atmospheric Hg | (Engle et al., 2010, Obrist et al., 2021) | 0.08 (0.05-0.12) |
| | Total | | 13.1 (7.7-19.5) | | This study | 0.52 (0.31-0.78) |
| | Export | Tidal export dissolved Hg | 0.7 | 100% marsh soil Hg | Wang and Obrist, 2022 | 0.03 |
| | | Tidal export particulate Hg | 5.6 | 100% marsh soil Hg | Wang and Obrist, 2023 | 0.22 |
| | | Wrack | 1.6 (1.4-1.7) | 100% marsh plants | This study | 0.06 (0.06-0.07) |
| | Total | | 7.9 (7.7 - 8.0) | | This study | 0.32 (0.31-0.32) |
| | **Net mass accumulation (estimated total deposition – total export)** | | **5.2 (0-11.5)** | | **This study** | **0.21 (0.0-0.46)** |
| | Internal Cycling | Green aboveground biomass* | 1.9±0.7 (0.7-2.6) | 35% soil Hg | This study | 0.08 (0.03-0.10) |
| | | Senesced aboveground biomass* | 1.1±0.6 (0.6-2.5) | 35% soil Hg | This study | 0.04 (0.02-0.10) |
| | | Total aboveground biomass | 3.1±1.1 (1.6-5.1) | 35% soil Hg | This study | 0.12 (0.06-0.20) |
| | | Roots and rhizomes | 58.6 (19.5-253.8) | 90% soil Hg | This study | 2.3 (0.8-10.2) |
| | | Herbivory | 0.5-2.4 | 100% marsh plants | This study | |
| | | Item | Hg mass (μg m⁻²) | Percent of Hg sources | Reference | Hg mass scaled to the marsh (kg)* |
| **Total Soil Hg Mass** | | **Soil Hg mass top 40 cm** | 25,575±14,409 (16,127-46,997) | | This study | 1,023±576 (645-1880) |

* Salt marsh area (vegetated): 40 km²

