# Peer review of "Above- and Belowground Plant Mercury Dynamics in a Salt Marsh"

_EGUsphere, 2023_

## Referee Comment (RC1)

Review of Manuscript preprint by Wang et al. 2023
Reviewer: Lena Wohlgemuth

The authors present interesting findings on mercury cycling in a salt marsh estuary based on Hg flux measurements and Hg isotope fingerprinting. Studying Hg dynamics in coastal ecosystems is necessary in order to better understand Hg export to coastal oceans. The study is comprehensive and well written. I support its publication with minor comments detailed below.

Line 25 – 34: The abstract could summarize increases in plant Hg during the growing season and the Hg mass balance in a more consolidated way. Just highlight the most relevant findings for the study.

Line 28/29: It should be made clear in the abstract, which values were not measured in this study, but taken from other studies.

Line 53: …accounting for 60% to 90% of total Hg inputs to soils

Line 53 – 59: Make the link to Hg. An uninformed reader might wonder why you go into detail of plant carbon assimilation in the introduction of a Hg paper.

Line 63: associated; ii) strictly speaking you did not quantify the transfer of Hg from aboveground biomass to soils. I think, i) comprehensively describes your objective and that you can delete ii).

Line 71: were

Line 78/79: Please check the species names again. Are they not called Sporobolus pumilus and Spartina alterniflora?

Line 84: Check the title

Section 2.2: For readers unfamiliar with this ecosystem, it is not obvious, that sampling salt marsh vegetation means sampling of a lot of senescent plants, i.e. that senescent plant material makes up a relevant proportion of total plant biomass throughout the growing season and not only in April of the following year. I think, this should be explained at some point in the paper, e.g. in Section 2.2 to avoid confusion.

Line 140: For the regression slope I assume that variability is presented by the standard error. Also, you could mention how you propagated standard deviations.

Line 145: …Hg concentrations (Fig. 1a)

Fig. 1: Where is data on Juncus gerardii? I suggest to use different colors for species and species communities.

Line 158 – 161: Can be moved to M&M

Line 165: Are numbers in brackets derived from pooled roots, rhizones, and detritus? This should be indicated here.

Line 161 – 173: I wonder if this section will improve by shortening it and highlighting only the most relevant concentration differences, that will be discussed later. All other values are already displayed in Fig. 2 or tabulated in the SI. This is a suggestion to make the concentration results section more engaging to the reader. You decide.

Line 183: …was 39% lower than total (live and senescent) biomass…

Fig. 2: The date when Hg concentrations = 0 was not determined, so don't indicate it by extending the regression line. Or did you do phenological observations to determine the beginning of the growing season?

Fig. 4b: Why are there two bars for April 2022? Please check.

Fig. 4c: Why are there different standard errors for April 2022?

Line 195: To be more precise and align the text to Fig. 6 please define "aboveground biomass" in this section once, e.g. "aboveground biomass (live and senescent marsh leaves)".

Line 215: For citing seasonality of Hg in forest foliage, please replace Wohlgemuth et al. 2022 with Wohlgemuth et al. 2020 (https://bg.copernicus.org/articles/17/6441/2020/).

Line 291: I redid the calculation in R (see code below) and the rounded model output was: f_atm = 0.33, f_root = 0.31, f_prep = 0.37. Almost the same values, but it might be worth checking the calculations again. Also, in the SI section on Hg isotope mixing model, please explicitly give the used endmember median values.

Line 290: I think a short explanation would be helpful for readers of how direct Hg uptake (translocation) from plant roots differs from precipitation Hg(II) deposition in this context, since precipitation water is also taken up by plant roots, which might confuse readers.

Line 371: I don't quite understand, how you estimated annual throughfall deposition. Did you derive it from Hg biomass at the end of the year (9.0 µg m$^{-2}$ yr$^{-1}$)? Theoretically, wash-off Hg could be a relatively constant value at every precipitation event over the growing season in a way, that wash-off Hg is independent from stomatal GEM or Hg root uptake. In fact, values presented in SI Table 5 do not support a clear increase of wash-off Hg over the growing season. Throughfall Hg is hard to quantify and maybe I misunderstand how you calculated 1.0 µg m$^{-2}$ yr$^{-1}$, but I think this merits an explanation in M&M.

Line 379: Give area in brackets

Line 392: Please mention the most relevant herbivores in this context (here or in Sect. 2.1). Readers are probably unfamiliar with the fauna of this ecosystem and this would help to understand why herbivory Hg is part of internal Hg cycling.

Fig. 6/Fig. S3: To me, the legends could be more intuitive. Is it possible to move the two legends from inside the plot panels to the right side, such that they apply to both Fig. 6a and 6b at first glance? Please note, that the isotope symbols representing organic soils and roots are almost identical by shape and color, same applies to rainfall and deep soil.

Fig. 7: This figure gives a good overview, however, I think you could improve it by clearly labelling, which arrows illustrate the fluxes used for the mass balance of this study and which arrows symbolize any possible Hg flux of the ecosystem. From my understanding, dashed deposition arrows represent Hg fluxes to aboveground biomass, non-dashed arrows represent Hg fluxes to the ground/soil pool (please define dashed/non-dashed in the caption). Therefore, I would extend the non-dashed deposition arrows of wet Hg(II), dry Hg(II), and GEM to the ground. For the deposition part of the mass balance you added up fluxes of wet and dry Hg(II) (both measured independent of aboveground biomass), throughfall Hg, and net GEM deposition determined from Hg accumulated in biomass and multiplied by the GEM percent contribution from the isotope mixing model (litterfall). So even though precipitation Hg(II) (dashed arrow) is taken up by aboveground tissues via the roots

(belowground dashed arrow would be more accurate), it is not part of the 4.1 µg m$^{-2}$ yr$^{-1}$ used for the mass balance, same is true for 2.1 µg m$^{-2}$ yr$^{-1}$ dry Hg(II). For GEM, you did not determine direct GEM deposition/re-emission to/from the ground, so what do dashed/non-dashed GEM arrows mean in this context?
Why is there a downward dashed flux arrow labelled "Aboveground biomass Hg uptake from soil 3.1", is this a mistake?

Table 1: Please make it clearer (e.g. with an asterisk), that Hg flux values (e.g. green/senescent biomass deposition of 3.7/2.1 µg m$^{-2}$ yr$^{-1}$) represent calculated percentages of measured values. I think, that this is not intuitive from the percent of Hg sources given in the next column for a reader, who only looks at this table without reading the text. The asterisk indicating the vegetated salt marsh area should only apply to the last column.

Line 434: herbivory internal cycling(?)

Line 433: Avoid repetitions of the abstract

Table S5: Please check the average value of estimated throughfall of S. alterniflora, it seems wrong. It is possible, that washed biomass samples (S. alterniflora in Oct-21 and Nov-21) are higher than respective unwashed samples due to measurement uncertainties and low Hg concentrations in wash-off, though don't give negative concentration values, but leave them out.

Section Summary and conclusion: You give all relevant fluxes, sources, and pools of the study, which is good. I wonder, if you could go a step further and bring this study in line with other studies on Hg input to coastal oceans, e.g. how this sink compares to other coastal sinks or input fluxes. Can you derive any implications from your findings for the ecosystem, e.g. in the introduction you mention, that the salt marsh is a Hg hotspot?

R code for checking Hg isotope mixing model

```
library(matlib)

**Isotope compositions:**
d202Hg_GEM <- -2.84
D200Hg_GEM <- -0.02
D199Hg_GEM <- -0.37
d202Hg_root <- -0.69
D200Hg_root <- 0.03
D199Hg_root <- 0.17
d202Hg_prep <- -0.3
D200Hg_prep <- 0.17
D199Hg_prep <- 0.4
D200Hg_veg <- median(c(0.11, 0.06, 0.07, 0.04))
d202Hg_veg <- median(c(-1.07, -1.61, -1.21, -1.29))
D199Hg_veg <- median(c(0.20, 0.43, 0.42, 0.32))

**ternary isotope mixing model**
A <- matrix(c(D200Hg_GEM, d202Hg_GEM, 1, D200Hg_root,
        d202Hg_root, 1, D200Hg_prep, d202Hg_prep, 1), 3, 3)
b <- c(D200Hg_veg, d202Hg_veg, 1)
showEqn(A, b)

Solve(A, b)
```

---

## Author Comment (AC1)

Review of Manuscript preprint by Wang et al. 2023 Reviewer:
Lena Wohlgemuth

The authors present interesting findings on mercury cycling in a salt marsh estuary based on Hg flux measurements and Hg isotope fingerprinting. Studying Hg dynamics in coastal ecosystems is necessary in order to better understand Hg export to coastal oceans. The study is comprehensive and well written. I support its publication with minor comments detailed below.

**Line 25 – 34: The abstract could summarize increases in plant Hg during the growing season and the Hg mass balance in a more consolidated way. Just highlight the most relevant findings for the study. lin**

**ANSWER:** We shortened this section and removed lines 25-29 to highlight the relevant finding.

**Line 28/29: It should be made clear in the abstract, which values were not measured in this study, but taken from other studies.**

**ANSWER:** We deleted the literature values in lines 28 and 29 based on the above question.

**Line 53: …accounting for 60% to 90% of total Hg inputs to soils**

**ANSWER:** Revised. (line 49 in current version)

**Line 53 – 59: Make the link to Hg. An uninformed reader might wonder why you go into detail of plant carbon assimilation in the introduction of a Hg paper.**

**ANSWER**: Thank you for pointing this out. We added a sentence to bridge the discussion between carbon and Hg assimilation in lines 55 to 56.

**Line 63: associated; ii) strictly speaking you did not quantify the transfer of Hg from aboveground biomass to soils. I think, i) comprehensively describes your objective and that you can delete ii).**

**ANSWER**: We shortened this sentence as suggested.

**Line 71: were**

**ANSWER:** Revised. (line 67 in current version)

**Line 78/79: Please check the species names again. Are they not called Sporobolus pumilus and Spartina alterniflora?**

**ANSWER**: We changed the species names accordingly.

**Line 84: Check the title**

**ANSWER**: Thank you, we changed the title to "**Sampling and Processing of Vegetation and Soil**". (line 87 in revised version)

**Section 2.2: For readers unfamiliar with this ecosystem, it is not obvious, that sampling salt marsh vegetation means sampling of a lot of senescent plants, i.e. that senescent plant material makes up a relevant proportion of total plant biomass throughout the growing season and not only in April of the following year. I think, this should be explained at some point in the paper, e.g. in Section 2.2 to avoid confusion.**

**ANSWER**: We clarified this information in lines 95 to 97.

**Line 140: For the regression slope I assume that variability is presented by the standard error. Also, you could mention how you propagated standard deviations.**

ANSWER: Thanks for the suggestion. I have revised the sentence in the manuscript in lines 153 to 154.

**Line 145: …Hg concentrations (Fig. 1a)**

**Fig. 1: Where is data on Juncus gerardii? I suggest to use different colors for species and species communities.**

ANSWER: Thank you for pointing out the issue. I have revised the graph Figure 1a.

**Line 158 – 161: Can be moved to M&M**

ANSWER: We moved this sentence and simplified the description to M&M in lines 103 to 105.

**Line 165: Are numbers in brackets derived from pooled roots, rhizomes, and detritus? This should be indicated here.**

ANSWER: In response to the comment below, we shortened this section and removed some of these numbers to avoid confusion (detailed numbers can be seen in Table S1 and in figures).

**Line 161 – 173: I wonder if this section will improve by shortening it and highlighting only the most relevant concentration differences, that will be discussed later. All other values are already displayed in Fig. 2 or tabulated in the SI. This is a suggestion to make the concentration results section more engaging to the reader. You decide.**

ANSWER: Thank you for your suggestion. We shortened this section to focus on more relevant information and refer to Figure 2 and Table S1.

**Line 183: …was 39% lower than total (live and senescent) biomass…**

ANSWER: corrected.

**Fig. 2: The date when Hg concentrations = 0 was not determined, so don't indicate it by extending the regression line. Or did you do phenological observations to determine the beginning of the growing season?**

ANSWER: We clarified in the figure legend that April data was extrapolated and set to zero based on phenological observations that showed no presence of live biomass.

**Fig. 4b: Why are there two bars for April 2022? Please check.**

ANSWER: We clarified in the figure legend that since there was no live biomass present in April 2022, senesced biomass equals total biomass at this point.

**Fig. 4c: Why are there different standard errors for April 2022?**

ANSWER: This was an error, we corrected this.

**Line 195: To be more precise and align the text to Fig. 6 please define "aboveground biomass" in this section once, e.g. "aboveground biomass (live and senescent marsh leaves)".**

ANSWER: We clarified that aboveground biomass means live and senesced marsh leaves. We also clarified that in this section we refer to live aboveground biomass only.

**Line 215: For citing seasonality of Hg in forest foliage, please replace Wohlgemuth et al. 2022 with Wohlgemuth et al. 2020 (https://bg.copernicus.org/articles/17/6441/2020/).**

ANSWER: Replaced. (line 221 of current version)

**Line 291: I redid the calculation in R (see code below) and the rounded model output was: f_atm = 0.33, f_root = 0.31, f_prep = 0.37. Almost the same values, but it might be worth checking the calculations again. Also, in the SI section on Hg isotope mixing model, please explicitly give the used endmember median values.**

ANSWER: Thank you for double checking. We reran the calculations with two Monte Carlo simulations and provide the best estimates as well as uncertainty ranges for these contributions. We further clarify the used endmember values in the calculation by referring to the supplementary document section "Hg Isotope Mixing Model" (Line 17-35). Additional text and a short section in SI has been provided to the Monte Carlo simulations to assess uncertainties.

**Line 290: I think a short explanation would be helpful for readers of how direct Hg uptake (translocation) from plant roots differs from precipitation Hg(II) deposition in this context, since precipitation water is also taken up by plant roots, which might confuse readers.**

ANSWER: We clarified that uptake of precipitation Hg includes root uptake of rainwater and possibly deposition to leaves, which we cannot distinguish in lines 63 to 64.

**Line 371: I don't quite understand, how you estimated annual throughfall deposition. Did you derive it from Hg biomass at the end of the year (9.0 µg m$^{-2}$ yr$^{-1}$)? Theoretically, wash-off Hg could be a relatively constant value at every precipitation event over the growing season in a way, that wash-off Hg is independent from stomatal GEM or Hg root uptake. In fact, values presented in SI Table 5 do not support a clear increase of wash-off Hg over the growing season. Throughfall Hg is hard to quantify and maybe I misunderstand how you calculated 1.0 µg m$^{-2}$ yr$^{-1}$, but I think this merits an explanation in M&M.**

ANSWER: Thanks for pointing out your confusion. We provided additional clarification in the main manuscript between lines 254 and 255. We clarified that in the absence of direct field throughfall measurements (which would be very challenging in a marsh system), the estimate is based on laboratory washing which showed on average, 11% of aboveground Hg can be washed off. We further state the limitation of this estimate and that is a very initial estimate in lines 352 to 353.

**Line 379: Give area in brackets**

ANSWER: The total salt marsh area of 40 km$^2$ is mentioned in section 2.1 Site Description in line 69 and Table 1 in line 761. To prevent any potential confusion, we've also included it in line 363 here.

**Line 392: Please mention the most relevant herbivores in this context (here or in Sect. 2.1). Readers are probably unfamiliar with the fauna of this ecosystem and this would help to understand why herbivory Hg is part of internal Hg cycling.**

ANSWER: We gave some examples of herbivores present in these ecosystems, including, marsh periwinkle and mummichog.

**Fig. 6/Fig. S3: To me, the legends could be more intuitive. Is it possible to move the two legends from inside the plot panels to the right side, such that they apply to both Fig. 6a and 6b at first**

glance? Please note, that the isotope symbols representing organic soils and roots are almost identical by shape and color, same applies to rainfall and deep soil.

ANSWER: Thanks for the suggestion. We changed the position of the legend as suggested.

**Fig. 7: This figure gives a good overview, however, I think you could improve it by clearly labelling, which arrows illustrate the fluxes used for the mass balance of this study and which arrows symbolize any possible Hg flux of the ecosystem. From my understanding, dashed deposition arrows represent Hg fluxes to aboveground biomass, non-dashed arrows represent Hg fluxes to the ground/soil pool (please define dashed/non-dashed in the caption). Therefore, I would extend the non-dashed deposition arrows of wet Hg(II), dry Hg(II), and GEM to the ground. For the deposition part of the mass balance you added up fluxes of wet and dry Hg(II) (both measured independent of aboveground biomass), throughfall Hg, and net GEM deposition determined from Hg accumulated in biomass and multiplied by the GEM percent contribution from the isotope mixing model (litterfall). So even though precipitation Hg(II) (dashed arrow) is taken up by aboveground tissues via the roots (belowground dashed arrow would be more accurate), it is not part of the 4.1 µg m$^{-2}$ yr$^{-1}$ used for the mass balance, same is true for 2.1 µg m$^{-2}$ yr$^{-1}$ dry Hg(II). For GEM, you did not determine direct GEM deposition/re-emission to/from the ground, so what do dashed/non-dashed GEM arrows mean in this context?**
**Why is there a downward dashed flux arrow labelled "Aboveground biomass Hg uptake from soil 3.1", is this a mistake?**

ANSWER: That you for your suggestions. We clarified that dashed arrows represent Hg fluxes related to the aboveground biomass dynamics and clarified this in the text. We changed the solid arrow for aboveground tissue deposition (including green, senesced, and throughfall) to a dashed line, and renamed this to "Aboveground tissue Hg uptake"). We also shifted the numbers for wet and deposition (4.1 and 2.1 µg m$^{-2}$ yr$^{-1}$) next to the solid lines to show these are non-vegetation depositions. Finally, we put a question mark next to the direct soil GEM emission/deposition since this was not measured here.

We intentionally didn't extend the non-dashed deposition arrows for wet Hg(II), dry Hg(II), and GEM via vegetation to the ground, because we don't need to partition respective deposition sources in this figure plus it would lead to difficult interpretations (as discussed by the reviewer, e.g., that we don't know the fraction of precipitation Hg take up directly on leaves versus transported from soil water). Finally, we removed the downward arrow of aboveground biomass Hg uptake, the reviewer is correct that this deposition is already represented in the aboveground tissue Hg uptake/turnover numbers and included in the total plant-derived deposition.

**Table 1: Please make it clearer (e.g. with an asterisk), that Hg flux values (e.g. green/senescent biomass deposition of 3.7/2.1 µg m$^{-2}$ yr$^{-1}$) represent calculated percentages of measured values. I think, that this is not intuitive from the percent of Hg sources given in the next column for a reader, who only looks at this table without reading the text. The asterisk indicating the vegetated salt marsh area should only apply to the last column.**

ANSWER: Thanks for your suggestion. We have removed the asterisks in the "Item" column. In the "Percent of Hg sources" column of Table 1, we have labeled their calculated percentages. Furthermore, the relevant information has been provided in the manuscript, specifically between lines 349 and 353.

**Line 434: herbivory internal cycling(?)**

**ANSWER**: Yes, the herbivory belongs to internal Hg cycling. In order to avoid confusion, I revised the sentence between lines 420 and 421.

**Line 433: Avoid repetitions of the abstract**

**ANSWER**: Thanks for your suggestion. We have revised the sentence between lines 418 and 421.

**Table S5: Please check the average value of estimated throughfall of S. alterniflora, it seems wrong. It is possible, that washed biomass samples (S. alterniflora in Oct-21 and Nov-21) are higher than respective unwashed samples due to measurement uncertainties and low Hg concentrations in washoff, though don't give negative concentration values, but leave them out.**

**ANSWER**: Thank you for bringing up this issue. We removed the "Estimated throughfall" values for individual months and now only provide an average value through the growing season to address this concern.

**Section Summary and conclusion: You give all relevant fluxes, sources, and pools of the study, which is good. I wonder, if you could go a step further and bring this study in line with other studies on Hg input to coastal oceans, e.g. how this sink compares to other coastal sinks or input fluxes. Can you derive any implications from your findings for the ecosystem, e.g. in the introduction you mention, that the salt marsh is a Hg hotspot?**

**ANSWER**: Thank you for your suggestion. We have added more discussion accordingly in lines 401 to 407.

R code for checking Hg isotope mixing model

```
library(matlib)

**Isotope compositions:**
d202Hg_GEM <- -2.84
D200Hg_GEM <- -0.02
D199Hg_GEM <- -0.37
d202Hg_root <- -0.69
D200Hg_root <- 0.03 D199Hg_root
<- 0.17 d202Hg_prep <- -0.3
D200Hg_prep <- 0.17

D199Hg_prep <- 0.4

D200Hg_veg <- median(c(0.11, 0.06, 0.07, 0.04)) d202Hg_veg
<- median(c(-1.07, -1.61, -1.21, -1.29))
D199Hg_veg <- median(c(0.20, 0.43, 0.42, 0.32))

**ternary isotope mixing model**
A <- matrix(c(D200Hg_GEM, d202Hg_GEM, 1, D200Hg_root,
d202Hg_root, 1, D200Hg_prep, d202Hg_prep, 1), 3, 3) b <-
c(D200Hg_veg, d202Hg_veg, 1) showEqn(A, b)

Solve(A, b)
```

---

## Author Comment (AC2)

Review of "Above- and Belowground Plant Mercury Dynamics in a Salt Marsh Estuary in Massachusetts, USA" by Wang et al. The authors carried out a systematic investigation on vegetation and soil in a salt marsh. It is important work to improve the understanding of Hg cycling in salt marsh area. However, the Hg isotopic mixing model exists critical mistakes. Firstly, some endmember isotopic signatures are lacking. The authors cited the global mean results, but I think this is not feasible. Citing results nearby the field site seems to be more convincing. Secondly, present model was built based on MDF and even-MIF. Plant uptake Hg from atmosphere or soil could cause significant negative MDF, so present model results are speculative. Thirdly, the authors cannot offer QA/QC about Hg isotopic measurements, and the error bars by measurement are lacking. In addition, the uncertainties of model estimations are not fully calculated.

**ANSWER:** There was a crucial misunderstanding of the reviewer in regard to the endmember mixing model, and this in part was due to a lack of clarification in our description. We DID EMPLOY both MDF and MIF fractionation in our calculations, and we DID EMPLOY the negative MDF induced by plant update from the atmosphere in the model. However, we failed to clarify that the negative MDF from plant uptake was used as an endmember by using the MDF signature from terrestrial foliage (with its strongly negative MDF values). Instead, the previous version suggested that directly used the atmospheric GEM signature (with positive MDF values). The clarification should resolve the concerns of the reviewer that model results are speculative. Finally, we added additional QA/QC data and an uncertainty analysis of the tertiary mixing model to the manuscript as described below.

In detail:

**Introduction: I suggest the authors should add one paragraph about Hg isotope, which was used to quantify the source of plant.**

**ANSWER:** Thank you for the suggestion. I have added a short section on the possible sources of Hg (and its isotope endmembers) between lines 62 and 64. More detailed information regarding the dominant end-member Hg sources are described in supplement document in section titled "Hg Isotope Mixing Model" (between lines 17 and 26).

**Line 20-21: Here peak aboveground Hg pool is 9.0 ug m-2, which is below that in November with 16.2 ug m-2. What differences?**

     **ANSWER:** We clarified this, but we think that the reviewer confused the units (concentration: $\mu g$ $kg^{-1}$ versus Hg pool: $\mu g$ $m^{-2}$).  The standing aboveground Hg pool is $9.0\pm3.3$ $\mu g$ $m^{-2}$ and represents the total Hg mass of both live and senesced aboveground biomass in November. The value 16.2 $\mu g$ $kg^{-1}$ (not $\mu g$ $m^{-2}$) is the Hg concentration of live aboveground biomass in November.

**Line 39: largest. How much is the global input flux?**

     **ANSWER:** We added this information in lines 37 and 38 with respective citations.

**Line 43-45: How about the anthropogenic pollution at present salt marshes?**

     **ANSWER:** We cannot say but revised the sentences here in lines 80 and 83. We also added a statement in the soils section that states that the small sink measured in this study is not consistent with

the large pool of Hg measured in soils in this marsh in lines 401 to 407. We discuss the potential for legacy/historical contamination from the industrial period in the region.

**Line 48-53: Cite more research papers, not only review.**

      **ANSWER:** We added specific research papers here, including lines 45 and 46 (Fisher and Wolfe, 2012; Iverfeldt, 1991; Rea et al., 1996; Zhou et al., 2021) and lines 47 and 48 (Fu et al., 2019; Jiskra et al., 2018; Obrist et al., 2018; Wang et al., 2019, 2022; Zhou et al., 2021; Zhou and Obrist, 2021).

      **Line 53: It is overestimated.**

      **ANSWER**: We refer to the range reported by several studies, we are not quite sure why this is considered overestimated.

**Line 53-59: I don't think it is related to the Hg cycling.**

      **ANSWER**: We added a sentence to clarify the linkage between NPP and Hg cycling in lines 55 and 56.

**Line 102-103: Were the soil samples ground by coffee grinders? I think such operation could cause cross contamination. How much mesh were sieved with? The soil homogeneity is very important.**

      **ANSWER**: We clarified that we used a soil shatter box (8530 Shatter-Box) to mill and homogenize soils samples with particle sizes less than 2mm. Coffee grinders were used to grind and homogenize plant samples only. Both the coffee grinders and the Shatter-Box were rinsed with Milli-Q water and dried with Kimwipes between samples. We clarified these methods in lines 107 to 110.

**Line 117-121: How about the Hg concentration in solution to isotopic measurement?**

      **ANSWER**: We added the requested information in lines 128 and 129.

**Line 127-130: Report the isotopic results of standard samples, i.e. NIST 1515 and MESS-4. I also cannot find those values in SI. QA/QC is important for Hg isotope measurement.**

      **ANSWER**: We have added the quality control results for standard samples in Table S4.

**Line 231-234: It is a good discussion about foliage accumulation by C3 and C4 plants. But it is not enough. I hope to get more discussion about the mechanism.**

ANSWER: We cannot go into details about mechanisms here as they are not well established and is beyond the scope of this paper. For example, the main study we know on this subject is from a 1983 study. We summarized the results of this study here (lines 239 and 240).

**Line 235-243: Such comparison in Hg concentration in different sites is tedious as too many factors controlled the Hg concentration.**

**ANSWER**: We agree, but we think it is important to refer to findings of other marsh plants studies here. We shortened the section, however, to make it less tedious.

**Line 259: How to identify the finer roots? <1mm or <2mm? Need more data to support it.**

**ANSWER**: We did not measure the exact diameter of *S. patens* (*S. pumilus*) roots. Instead, we labeled them 'finer roots' in comparison to *S. afterniflora* (*S. alterniflorus*), based on our personal observations. We clarified this in in brackets (at line 261 of the current paper version).

**Line 277: Recent study has demonstrated the inconsistent MIF between foliage and atmosphere.**

**ANSWER**: Many previous studies have indicated similar signatures of MIF between foliage and atmosphere. However, we do now mention that one recent study reported inconsistent MIF between foliage and the atmosphere in lines 279.

**Line 281-282: The authors should compare the isotopic data with the reported grass isotopic values, rather than all the plant values. As you said, the Hg assimilation may exist great difference between C3 and C4 plant.**

**ANSWER**: Ideally, this should be done, but to our knowledge there simply are not sufficient studies on Hg isotopes in grasses and differences between C3 and C4 plants. Hence, our best approach is to use the wider data sets from upland plants.

**Line 283-285: It is speculatively. I do not agree with it.**

**ANSWER**: We have deleted this sentence .

**Line 288: How to build the model based on MDF? As we know, the plant uptake Hg could induce significant MDF. I do not believe this model results if it is based on MDF.**

**ANSWER**: As written in our response to the first statement, there is a confusion and we indeed used the significant MDF that occur by plant uptake of atmospheric GEM. The end-member in the ternary isotope mixing model we used to calculate MDF is based on Hg isotopic signatures ***of upland foliage with negative MDF values, not atmospheric GEM as we originally wrote.*** We have clarified this in the manuscript in lines 289 and 290.

Detailed information about the mixing model can be found in the supplementary document (between lines 17 and 26). In summary, the three dominant end-member Hg sources (medians) include local marsh plants roots, terrestrial upland foliage (used to represent atmospheric GEM), and precipitation. The three calculation equations are also provided in the supplementary document. Hopefully, this information proves helpful.

**Line 291-292: How about the uncertainty of this estimation?**

**ANSWER**: We now provide a detailed assessment of the uncertainties of our tertiary mixing model. Due to the low sample numbers and reported analytical uncertainty, the uncertainties are

substantial. However, two different methods to assess the uncertainties both show similar mean values of contributions. We clarified the uncertainty analysis in the section "Hg Isotope Mixing Model" in SI between lines 30 and 35.

**Line 291-303: I do not believe the model results. Briefly, all the MDF values are more positive than that in measured vegetation results in Figure 6 a). I cannot think this mixing model can be solved in present pattern. Further, I suggest to add error bars (2SD) in Figure 6.**

**ANSWER**: We added error bars to Figure 6, but their impact may not be immediately apparent since we only have two samples analyzed in duplicate with error bars. We refer to our response above about the MDF, this clarification should address the concerns of the reviewer.

**Line 304-312: This paragraph is speculatively. Not all the slope of ~1 suggested the inorganic Hg photoreduction.**

**ANSWER**: We deleted this paragraph to avoid confusion and removed the discussion points.

**Line 414-422: Turnover flux is an interesting discussion. But I think such flux is overestimated.**

**ANSWER**: We are not sure why the reviewer thinks it is an overestimate, we did our best to estimate the potential magnitude of these different component fluxes based on detailed turnover estimation of marsh ecosystem studies. As discussed, please note that the large turnover flux is largely attributed to the substantial Hg mass present in plant roots and marsh soils and as such constitutes an internal system recycling of Hg between plant-bound and soil Hg.

**Line 745-749: Please cite the original research papers, not the review paper**.

**ANSWER**: For this figure, we used and overlaid data from a figure from this review paper so citing this study here seems appropriate.

**Line 752: Dry HgII deposition flux with 2.1. How to estimate it? Add the references.**

**ANSWER**: The discussion regarding the dry HgII deposition and relevant references can be found in Section '4.3.1 Aboveground' (line 346-348). Additionally, the estimated dry HgII deposition referenced is also provided in Table 1. (Engle et al., 2010, Obrist et al., 2021)